# Correlation between Caries, Body Mass Index and Occlusion in an Italian Pediatric Patients Sample: A Transverse Observational Study

**DOI:** 10.3390/ijerph17092994

**Published:** 2020-04-26

**Authors:** Angela Militi, Riccardo Nucera, Ludovica Ciraolo, Angela Alibrandi, Rosamaria Fastuca, Roberto Lo Giudice, Marco Portelli

**Affiliations:** 1Department of Biomedical Sciences, Dentistry and Morphological and Functional Imaging, University of Messina, Via Consolare Valeria 1, 98122 Messina, Italy; 2Department of Economics, University of Messina, 98122 Messina, Italy; 3Department of Medicine and Surgery, School of Medicine, University of Insubria, Via G. Piatti 10, 21100 Varese, Italy

**Keywords:** BMI, malocclusion, caries

## Abstract

Background: The aim of this study was to evaluate the correlation between caries, body mass index (BMI) and occlusion in a sample of pediatric patients. Methods: The study group included 127 patients (72 female, 55 male) aged between 6 and 16 years (mean age 10.2) and selected between January and June 2019 at the Department of Pediatric Dentistry, University of Messina. Caries incidence was evaluated using the decayed, missing and filled teeth (DMFT) index. On the basis of BMI values, using a table adjusted for age and gender, patients were grouped into four categories (underweight, normal weight, risk of overweight, overweight). Results: There was no significant correlation between BMI and DMFT in the whole sample. The study of the correlation between BMI and DMFT in patients with different types of malocclusion showed a significant inverse correlation for patients affected by II class and deepbite malocclusion. Conclusions: The incidence of caries does not seem to be significantly related to BMI and occlusal patterns, but it decreases with increasing age.

## 1. Introduction

Over the past two decades, overweight and obesity have increased among children and adolescents around the world, representing a public health concern. A study performed by the World Health Organization analyzed the weight and height of around 130 million people, including 31.5 million people aged between 5 and 19 years. The aim of this study was to observe how the levels of body mass index (BMI) and obesity have changed in the last 40 years; the results of this analysis showed a substantial increase in the body weight of growing subjects [1]. The varying prevalence of obesity according to socioeconomic level is influenced by two main risk factors: insufficient physical activity and unhealthy diet [2,3]. Oral health and obesity share common risk factors, such as genetic and socioeconomic factors and eating disorders [4,5]. Caries significantly affects the state of oral health because decayed teeth, if not treated, in addition to a progressive destruction of teeth, can produce parodontal problems, halitosis, occlusal and esthetic effects, etc. Dental caries is a multi-factorial disease and affects most of the world population. It is the primary cause of oral pain and tooth loss. It is considered one of the major health-related problems in young children and one of the most prevalent oral diseases [6]. For epidemiological purposes, the most frequent index used to evaluate the intensity of caries is the number of decayed, missing and filled teeth—the so-called DMFT index—which is used worldwide. Several clinical studies have investigated the relation between obesity and dental caries by the mean of BMI and DMFT. Different studies reported an association between the increase in body weight status in children and the development of caries [7,8]. Other studies instead revealed an increased incidence of caries in underweight patients [9,10]. Other authors, such as Peng and De Jong, have not found evidence of an association between the two variables [11,12]. Different authors have evaluated the association between caries and obesity considering also various third correlating factors, such as socioeconomic conditions [13,14,15,16] demographic items [17], oral hygiene [18], periodontal status [19], general clinical history [20], salivar characteristics [21] or physical activity [22]. However, no study to date has assessed subject occlusion as the third factor. However, occlusion can play an important role in caries incidence, because some malocclusion can lead to areas of low detergency, both salivary and mechanical. Our hypothesis is that a correlation exists between the incidence of caries, weight and occlusal characteristics of growing subjects. The aim of this transverse observational study was to evaluate the correlation between caries, body mass index (BMI) and occlusion [23] at pediatric age. 

## 2. Materials and Methods 

This observational study was performed between January and June 2019 at the University of Messina, Department of Pediatric Dentistry, University Hospital “G. Martino”, Messina. The study included 127 patients (72 female, 55 male) aged between 6 and 16 years (mean age 10.2), all of whom came from oriental Sicily and south Calabria, with their parents being natives of the same geographical areas. Clinical examination was performed by the same experienced operator, using predefined data collection forms, after acquiring the informed consent and data from the parents or legal guardians of the child. The protocol was reviewed and approved by the Ethical Committee (Approval No. 437, 02 October 2018), and the procedures followed adhered to the World Medical Organization Declaration of Helsinki. Unhealthy patients and those suffering from systematic disease, undertaking any pharmacological therapy or affected by dental diseases [24,25,26,27,28,29] were excluded from the study. Both patients’ weight (kg) and height (cm) were registered twice for each patient in order to prevent any measurement error [30]. These values were used to calculate the body mass index, using the formula BMI = kg/m^2^. The BMI values for age and gender were evaluated taking into consideration the graphs developed by the Center for Disease Control (CDC) 2000 standards [31]. According to these curves, the subjects were classified into four weight groups.

Underweight: BMI by age below the fifth percentile;Normal: BMI by age greater than or equal to the fifth percentile and less than the 85th percentile;At risk of being overweight: BMI by age greater than or equal to the 85th percentile and less than the 95th percentile;Overweight: BMI by age greater than or equal to the 95th percentile.

The sample size was established assuming an effect size of 0.25 for correlation between BMI and DMF [32], a two-sided significance level of 5% and a power of 80%. Based on these assumptions, a minimum number of 120 patients was necessary to ensure an adequate statistical power. Our sample was composed of 123 subjects; therefore, it reached a power of 81.1%. Power and sample size calculation was performed using G-power software (3.1.9.4 version, HHU, Dusseldorf, Germany). The incidence of caries was evaluated with the DMFT index recording the number of decayed, missing and filled teeth. Patients’ clinical examination was performed following the guidelines to prevent infections. Occlusion was evaluated using the anteroposterior relationships of the maxillary and mandibular first molars and canine in maximum intercuspation according to Angle’s classification. Dental analysis also included the overbite evaluation, which is the vertical relationship of the upper and lower incisors. This was recorded as increased when the maxillary central incisors covered the mandibular central incisors by more than 3 mm. An anterior open bite was recorded when the incisal edges of the maxillary incisors did not overlap the incisal edges of the mandibular incisors. Overbite was measured using a ruler which was able to measure the vertical distance between the superior and inferior incisal margin in occlusion. The evaluation was repeated in 20 randomly selected patients to assess the intra-operator reliability. Systematic and random errors were calculated comparing the first and second measurements with dependent t-tests and Dahlberg’s formula at a significance level of *p* < 0.05. All measurement error coefficients were found to be adequate for the appropriate reproducibility of the study. 

## 3. Statistical Analysis 

Categorical variables were expressed as the absolute frequency and percentage while the numerical variables were expressed as the mean and standard deviation. The Kolmogorov–Smirnov test was applied to verify the variable distribution’s normality. This test allowed us to determine the statistically significant deviation of normality for the three variables of age, BMI and DMFT; consequently, we decided to use a non-parametric approach for the statistical analysis of the data. Spearman’s test was used to calculate the statistical dependence between the rankings of the two variables BMI and DMFT. The same analysis was conducted for each malocclusion class and overbite type. Lastly, these values were correlated to age. The significance level set for the analysis was α = 0.050; thus, for the two-sided CI, only *p*-values inferior to 0.050 were considered as statistically significant. The statistical software used was SPSS for Windows, version 22.0(IBM, New York, NY, United States).

## 4. Results

After calculating the BMI for each subject, the data were catalogued in four groups: underweight, normal weight, risk of overweight and overweight. The absolute frequency and percentage were calculated as recorded in Table 1.

The same analysis was used to calculate the frequency and percentage of malocclusion types (Table 2).

For each overbite type, the percentage and frequency were calculated counting non-valuable subjects in Table 3 and excluding them in Table 4.

Descriptive statistics in terms of the mean value and standard deviation for the numerical variables of age, BMI and DMFT are reported in Table 5.

The Kolmogorov–Smirnov (Table 6) normality test showed that the distribution of the data of BMI and DMFT values was not normal, so it was decided to apply non parametric tests to conduct the statistical evaluation.

The correlation between BMI and DMFT was calculated in a sample of 127 subjects using the Rho Spearman correlation index. The correlation between the two variables is not positive (Table 7).

The correlation between BMI and DMF was also calculated by subdividing subjects into four groups: underweight, normal weight, risk of overweight and overweight. There is a significant (two-tailed asymptomatic significance) inverse relationship for normal weight patients; i.e., when BMI rises, DMF decreases (Table 8). 

In underweight (Table 9), risk of overweight (Table 10) and overweight (Table 11) subjects, there were no statistically significant results.

The Spearman index was also used in subjects with Class I and Class III malocclusion (Table 12 and Table 13) to evaluate the correlation between BMI and DMFT. This result is not statistically significant at *p* > 0.050.

In Class II patients, the correlation was inverse (coefficient = −0.377) and was statistically significant (*p*-value < 0.050). This means that when BMI increases, DMFT decreases (Table 14).

The statistical study also included an overbite evaluation of the samples analyzing the variables BMI and DMFT. In deepbite subjects, the correlation between the two variables was almost statistically significant (i.e., the *p*-value was near the 0.050 level); therefore, there is a weakly negative correlation between BMI and DMFT (Table 15).

On the other hand, no significant correlation was shown for openbite (Table 16) and normal occlusion (Table 17) patients between the two variables.

Applying the Rho Spearman coefficient to the correlation between BMI, DMFT and age (Table 18), it was found that with increasing age, BMI also rises in a significant way (*p* > 0.001). This test shows that DMFT decreases with increasing age, showing that the incidence of caries decreases with increasing age. Demographic and clinical variables of patients according to gender are reported in Table 19.

## 5. Discussion

This transverse observational study has been performed to evaluate the correlation between caries, body mass index (BMI) and occlusion in a sample of pediatric patients; the correlation between obesity and caries allowed us to verify the relations of interdependency between the variables BMI and DMFT for each class of malocclusion and each type of overbite. The correlation between BMI and DMFT in our sample revealed a negative coefficient with the data available in the literature. Normal weight patients showed a significant inverse correlation of the two variables, while in overweight and risk of overweight patients, the statistical correlation was not statistically significant. On the basis of the results of the present study, BMI does not appear to be significantly related to caries incidence. The study of the correlation between BMI and DMFT in patients with each type of malocclusion showed a significant inverse correlation only in class II and deepbite patients. For all other types of malocclusion and overbite, the correlation was not statistically significant. On the basis of the results of the present study, molar occlusion and bite depth patterns do not seem to be significantly related to caries incidence; however, further studies are necessary to evaluate a possible correlation between the degree of dental crowding and caries incidence. The results of the present study agree with the finding of a systematic review performed by Paisi et al. [33]. At the end of the review process, only seven studies have been considered: two of them—conducted in an Indian [32] and Saudi Arabian [34] population sample—showed a positive correlation between the two variables, while the other five studies did not find any association between caries and BMI [35,36,37,38,39]. Honne [33] found a positive correlation between BMI, decayed teeth (DT) and the sum of decayed, missing and filled teeth (DMFT) in 463 adolescents aged between 13 and 15. The study also proved that the risk of caries in overweight subjects was 3.68 times higher than normal or underweight subjects. Sakeenabi [34] examined a sample of 1550 children, and he found that in 6-year-old overweight patients, the risk of caries was 1.92 times higher than normal weight patients. The risk of caries in 13-year-old overweight patients was 1.68 times higher than normal weight subjects. These two studies found different results than ours, because they revealed a positive correlation between BMI and DMF in overweight patients. The studies by Honne and Sekeenabi, performed in an Indian population sample, found a positive correlation between caries and BMI; their results may depend on the number of subjects enrolled in the studies and their geographical origin. In the study performed by Peng et al. [35], no correlation was found between caries and weight; however, this study is not comparable to ours, because BMI has been evaluated in a different way. In the study performed by Dye et al. [36] in children aged between 2 and 5 years old, the aim was to evaluate the correlation between caries in deciduous teeth and eating habits. However, in this study, which focused only on milk teeth, the DMF index was not related to the BMI of children but to the daily fruit consumption and quantity of milk consumed; no correlation was evaluated with the weight characteristics of the child. In the study performed by Hong et al. [36], the correlation between BMI and caries in 1507 children aged between 2 and 6 years old was evaluated; the results showed a statistically significant association between caries and obesity only in 60–72-month-old subjects; the remaining sample did not show any significant correlation between the two variables. The study conducted by Jürgensen et al. [38] evaluated a sample of 624 subjects at 12 years old and did not highlight any association between BMI and oral health or common risk factors; the evaluation of the data consisted in the clinical registration of caries, parodontal state and dental trauma. Tramini et al. [39] examined a sample of subjects aged between 11 and 12 years old; in this study, the authors evaluated not only DMF and BMI but also the preference and sensibility to sweet and bitter. No statistically significant association was found between caries and weight. According to the data found in the systematic review, this study proves that it is not possible to find a positive correlation between the variables BMI and DMFT. However, interesting data came to light that lead to an inverse correlation between caries and obesity in children and adolescents. According to the result of this observational transverse study, is possible to state that the incidence of caries decreases with increasing age; moreover, in younger patients affected by weight disorders, caries incidence was found to be higher in comparison to adolescents. These data may be related to the increase of the oral hygiene level and to the better eating habits that often occur with children’s growth. Obesity and caries have several common risk factors, and in pediatric patients, they require a multidisciplinary approach both by medical and dental clinicians. For a more accurate and specific analysis, the next study should also take into account a better evaluation of oral hygiene methods and plaque indexes using parodontal probing as well. It is also necessary to further analyze the BMI evaluation considering eating habits and the DMF index considering respiratory and functional habits [40,41,42,43,44,45,46,47]. The clinical examination could be also associated with radiological investigations with specific low-dose protocols [41,48,49,50] in order to better evaluate any caries which are not clinically evident and to assess occlusal patterns not only in the context of the dental relationship but also considering skeletal factors.

## 6. Conclusions

According to the results of the present study, no statistical correlation has been found between BMI and DMFT, but a decrease of caries incidence has been observed with increasing age. A significant inverse correlation between BMI and DMFT in patients with each type of malocclusion was shown only in Class II and deepbite patients; occlusal patterns do not seem to be related to caries incidence. However, more evidence is needed in order to evaluate the correlations between BMI, malocclusion and caries incidence.

## Figures and Tables

**Table 1 ijerph-17-02994-t001:** Frequency and percentage of the sample, divided according to the weight curves.

BMI	Frequency	Percentage
Underweight	11	8.7%
Normal Weight	60	47.2%
Risk of overweight	28	22.0%
Overweight	28	22%
Total	127	100%

**Table 2 ijerph-17-02994-t002:** Frequency and percentage of the sample divided according to the molar relationship.

Malocclusion	Frequency	Percentage
I Class	41	32.3%
II Class	42	33.1%
III Class	44	34.6%
Total	127	100%

**Table 3 ijerph-17-02994-t003:** Frequency and percentage of patients divided according to the depth of the bite.

Overbite	Frequency	Percentage
Deepbite	39	30.7%
Openbite	22	17.3%
Normal occlusion	60	47.2%
Not evaluable	6	4.7%
Total	127	100%

**Table 4 ijerph-17-02994-t004:** Frequency and percentage of patients divided according to the depth of the bite, excluding non-valuable subjects.

Overbite	Frequency	Percentage
Deepbite	39	32.2%
Openbite	22	18.2%
Normal occlusion	60	49.6%
Total	121	100%

**Table 5 ijerph-17-02994-t005:** Mean value and standard deviation of the age, body mass index (BMI) and DMFT index of the sample.

	Mean	Standard Deviation
Age	9.87	3.55
BMI	19.69	4.88
DMF	3.28	3.17

**Table 6 ijerph-17-02994-t006:** Kolmogorov–Smirnov test for the sample.

	BMI	DMF
Z Kolmogorov–Smirnov	1.268	2.033
Sig. Asint. two-sided CI	0.039	0.001

**Table 7 ijerph-17-02994-t007:** Data related to the correlation between the BMI and DMF evaluated in the whole sample.

	BMI	DMF
BMI coefficient of correlation Sig. (two-sided CI) *N*	1.000 127	−0.155 0.82 127
DMF coefficient of correlation Sig. (two-sided CI) *N*	−0.155 0.082 127	1.000 127

**Table 8 ijerph-17-02994-t008:** Data related to the correlation between BMI and DMF evaluated in normal weight patients.

	BMI	DMF
BMI coefficient of correlation Sig. (two-tail) *N*	1.000 60	−0.290 * 0.024 60
DMF coefficient of correlation Sig. (two-tail) *N*	−0.290 * 0.024 60	1.000 60

* The correlation is significant at the 0.05 level (two-tail).

**Table 9 ijerph-17-02994-t009:** Data related to the correlation between BMI and DMF evaluated in underweight patients.

	BMI	DMF
BMI coefficient of correlation Sig. (two-tail) *N*	1.000 11	0.234 0.489 11
DMF coefficient of correlation Sig. (two-tail) *N*	0.234 0.489 11	1.000 11

**Table 10 ijerph-17-02994-t010:** Data related to the correlation between BMI and DMF evaluated in risk of overweight patients.

	BMI	DMF
BMI coefficient of correlation Sig. (two-tail) *N*	1.000 28	−0.289 0.136 28
DMF Coefficient of Correlation Sig. (two-tail) *N*	−0.289 0.136 28	1.000 28

**Table 11 ijerph-17-02994-t011:** Data related to the correlation between BMI and DMF evaluated in overweight patients.

	BMI	DMF
BMI coefficient of correlation Sig. (two-tail) *N*	1.000 28	−0.018 0.929 28
DMF coefficient of correlation Sig. (two-tail) *N*	−0.018 0.929 28	1.000 28

**Table 12 ijerph-17-02994-t012:** Data related to the correlation between BMI and DMF in patients with a Class I molar relationship.

	BMI	DMF
BMI coefficient of correlation Sig. (two-tail) *N*	1.000 41	−0.193 0.227 41
DMF coefficient of correlation Sig. (two-tail) *N*	−0.193 0.227 41	1.000 41

**Table 13 ijerph-17-02994-t013:** Data related to the correlation between BMI and DMF in patients with a Class III molar relationship.

	BMI	DMF
BMI coefficient of correlation Sig. (two-tail) *N*	1.000 44	0.095 0.538 44
DMF coefficient of correlation Sig. (two-tail) *N*	0.095 0.538 44	1.000 44

**Table 14 ijerph-17-02994-t014:** Data related to the correlation between BMI and DMF in patients with a Class II molar relationship.

	BMI	DMF
BMI coefficient of correlation Sig. (two-tail) *N*	1.000 42	−0.377 * 0.014 42
DMF coefficient of correlation Sig. (two-tail) *N*	−0.377 * 0.014 42	1.000 42

* The correlation is significant at the 0.05 level (two-code).

**Table 15 ijerph-17-02994-t015:** Data related to the correlation between BMI and DMF in patients affected by deepbite.

	BMI	DMF
BMI coefficient of correlation Sig. (two-tail) *N*	1.000 39	−0.303 0.061 39
DMF coefficient of correlation Sig. (two-tail) *N*	−0.303 0.061 39	1.000 39

**Table 16 ijerph-17-02994-t016:** Data related to the correlation between BMI and DMF in patients affected by openbite.

	BMI	DMF
BMI coefficient of correlation Sig. (two-tail) *N*	1.000 22	−0.311 0.159 22
DMF coefficient of correlation Sig. (two-tail) *N*	−0.311 0.159 22	1.000 22

**Table 17 ijerph-17-02994-t017:** Data related to the correlation between BMI and DMF in normal occlusion patients.

	BMI	DMF
BMI coefficient of correlation Sig. (two-tail) *N*	1.000 60	−0.111 0.396 60
DMF coefficient of correlation Sig. (two-tail) *N*	−0.111 0.396 60	1.000 60

**Table 18 ijerph-17-02994-t018:** Data related to the correlation between BMI, DMF and age for the whole sample.

	BMI	DMF	Age
BMI coefficient of correlation Sig. (two-tail) *N*	1.000 127	−0.155 0.082 127	0.475 ** 0.00 127
DMF coefficient of correlation Sig. (two-tail) *N*	−0.155 0.082 127	1.000 127	−0.306 ** 0.00 127
ETA’ coefficient of correlation Sig. (two-tail) *N*	0.475 ** 0.000 127	−0.306 ** 0.000 127	1.000 127

** Correlation is significant at the level 0.01 (two-tail).

**Table 19 ijerph-17-02994-t019:** Demographic and clinical variables of patients according to gender.

Variables	Male	Female	*p* Value
**BMI**	19.3 ± 4.6	19.9 ± 5.1	0.744
**DMF**	3.5 ± 3.2	3.1 ± 3	0.501
**Age**	9.6 ± 3.5	10 ± 3.6	0.535
Normal weight Risk of overweight Overweight Underweight	47.3% 23.6% 20.0% 9.1%	47.2% 20.8% 23.6% 8.3%	0.957
**Malocclusion** I Class II Class III Class	30.9% 30.9% 38.2%	33.3% 34.7% 31.9%	0.762
**Overbite** Deepbite Openbite Normal occlusion Not evaluable	32.7% 14.5% 49.1% 3.6%	29.2% 19.4% 45.8% 5.6%	0.832

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
