# Peer review of "Correlation between Caries, Body Mass Index and Occlusion in an Italian Pediatric Patients Sample: A Transverse Observational Study"

_ijerph, 2020, doi:10.3390/ijerph17092994_

Round 1
Reviewer 1 Report
This is an interesting paper in scope although there are a considerable number of spelling, grammar and English errors. The intro and discussion are well organized. The methods unfortunately, are poorly organized and not very clear. The study population must be more explicitly described. The age range is very broad, and may have had a significant impact on the results, this may also be a limitation that should be addressed in the discussion. I recommend that the authors format the 1st unedited methods paragraph into clear subsections. The results are very poorly presented. There is very little narrative or reference to the tables. The tables themselves are too numerous, confusing with limited formatting. The results should be described in detail with references made to a discrete number of illustrative tables or figures. The authors should also include analysis on the effect of sex based on the mixed study population demographic. Table legends should be clear and descriptive. An acknowledgment of on study limitations should be included in the discussion. The use of abbreviations and acronyms should be avoided unless they are described first in the text (e.g. Sig 2-code). With adequate descriptions and explanations, the study is of significant value and would warrant publication.
Author Response
Dear Reviewer
Thank you so much for the corrections suggested.
Please see the attachment.
My best regards

Reviewer 2 Report
Review ijerph-769038
Thank you for your submission.The relation between caries, body mass index and occlusion in pediatric patients is an interesting topic, that requires further investigation. Please, consider the following points to improve your manuscript:
Please, revise the English grammar of the text. There are some mistakes and errors with the scientific language. Examples:
Introduction:
Page 2, line 52:
This index it's easy to comprehend and it does not depend on therapy
Page 9, line 195:
this demostrate that the number of caries is reduced with aging.
Page 10, line 212: letterature.
Page 11, line 233:
Howevere
Page 11, conclusions:
hygene; invetsigations; Singificance
The study was performed in the past. Authors start the materials and methods section with past tense and then the text continues in the present. Please, be consistent with verbal tenses.
Materials and methods:
Page 2, line 64: was performed
Line 67: has been performed
This sentence is redundant, Angle’s classification is auto-defined by itself:
Materials and methods:
Page 2, lines 79-81:
Occlusion has been evaluated using the anteroposterior relationships of the maxillary and mandibular first molars and canine in maximum intercuspation according to Angle’s classification.
This is a test-retest evaluation. Please, define the evaluation of the intra-operator reliability in this way.
Materials and methods:
Page 3, lines 87:
The evaluation has been repeated in 20 randomly selected patients, to assess the intra-operator reliability.
The BMI values assigned to the different groups have to be defined previously in the Materials and methods section:
Results:
Page 3, Line 104
4 groups: underweight, normal weight, risk of overweight and overweight.
Is not possible that in the general population authors obtain the same approximately number of patients with the different types of Angle occlusion. It is not a major concern for the purposes of this study, but this seem to be a convenience sample that should be explained in the Materials and Methods section.
Page 3 Table 2:
The total number of patients in Table 4 continues to be 127, authors have distributed the 6 Not evaluable patients in the other three groups. The total number of patients in that table is not 121.
Table 4, page 4, line 117:
Total 121 100%
This sentence is an interpretation of the results. It must be moved to the discussion section.
Page 7, line 161:
This means that when BMI increases DMFT decreases
Conclusions must show strictly the response to the author’s objectives. The major part of the conclusions section offers information that should be moved to the discussion section.
Conclusions:
Page 11, line 254:
This data may be related to the increase of oral hygiene level and to the better eating habits that often occur with children growth. Obesity and caries have several common risk factors and in pediatric patients they require a multidisciplinar approach by both medical and dental clinicians. For a more accurate and specific analysis the next study should also take into account a better evaluation of oral hygene methods and plaque index using parodontal probing. It is also necessary to further analyze the BMI evaluation considering eating habits, and DMF index considering respiratory and functional habits [39-46]. Clinical examination could be also associated to radiological invetsigations with specific low-dose protocol [47-49] , in order to better evaluate any caries not clinically evident. To increase the value of the study and to obtain a much higher statistically singificance it should be used a greater number of subjects.
Author Response

(The authors gave the same response as above.)

Reviewer 3 Report
Review report:
Title: Correlation between caries, body mass index and occlusion in an Italian pediatric patients’ sample: a transverse observational study
This manuscript while conceptually offers an addition to the literature, it however has several drawbacks that makes it not suitable for publication in its current format. This review is meant to point out issues that may improve this manuscript and perhaps increase the robustness of the study.
The manuscript requires proof reading, evident by several typos and unclear sentences that makes it difficult for the readers to understand. Below are examples of that:
e.g. 1, Line 40, it reads “aged between 5 and 19 years; whit the aim to observe how the levels of body mass index (BMI) and”
e.g. 2, line 45, it reads: “diriment role in oral health because decayedteeth, if not treated, can cause the activation of”
e.g. 3, line 248, it reads: “to the data found in the sistematic review the study proves that it is not possible to find a positive”
Abstract:
Line 32 and 33, under conclusions,
Comment: The sentence needs to be re-structures since it is difficult to understand.
Introduction:
Lines 44,45 and 46, it reads “Caries have a diriment role in oral health because decayedteeth, if not treated, can cause the activation of infectious process in gums and parodontal tissue, halitosis, occlusal and aestethic problems”.
Comment: This sentence does not make a strong argument and I suggest that it is to be rewritten for clarity.
Lines 51 and 52, It reads: “This index it's easy to comprehend and it does not 51 depend on therapy”
Comment: I suggest to re-write this sentence for clarity.
Lines 54 and 55, it reads: “Other studies revealed instead, that lower weight status is associated with a greater experience of caries”
Comment: I suggest to re-write this sentence for clarity.
Lines 57, 58, 59, It reads: “Different studies conducted on the association between caries and obesity have considered a third correlating factor besides BMI and DMF, such as the socioeconomic and demographic factor, oral hygiene.
Comment: I suggest to re-write this sentence for clarity.
The aim of the study, on lines 61 and 62: it reads: “The aim of this transverse observational study was to evaluate the correlation between caries, body mass index (BMI) and occlusion [23]”
Comment: The reference number 23 that is used in the aim seem to be a case report!. Why would the aim of the study be referenced to a case report? That need to be clarified or omitted since it unusual.
There is no clear hypothesis or null hypothesis identified or eluded to in this manuscript.
While the correlation between caries and body mass index has been investigated and reported, but the rational of the correlation with occlusion or malocclusion is not made clear in this study. This has to be made clear and as I mentioned earlier, a hypothesis to this proposed correlation or association is not identified.
Materials and Methods:
Comment: This section needs to be rewritten to make it clearer to the readers.
The study was made of a sample of 127 participants. Why 127? What’s the power calculation? If the number was more would that have resulted in a different data? Hence, power calculation would be important.
Discussion:
The structure of the study resulted in the section to be less than ideal. The comparison with other studies may not be as strong, given the small sample size in this study against the studies that the authors mentioned.
In addition, the correlation with occlusion has not been discussed and no interpretation to findings been offered, in relation to occlusion or malocclusion.
Conclusion:
This section should relate to conclusions of the findings of the study rather than interpretations related to the findings, which should be kept at the discussion section. It should be succinate and to the point of the study rather than discussion of other issues, like oral hygiene.
Author Response

(The authors gave the same response as above.)

Round 2
Reviewer 2 Report
Thank you for the revisions, authors have done a good job improving the manuscript.
Author Response
Dear Reviewer
Thank you so much for your valuable time and usefull contribution. We appreciate the imputs received from you, and thank to your suggestions we have improved the manuscript entitled “Correlation between caries, body mass index and occlusion in an Italian pediatric patients sample: a transverse observational study”.
We are pleased to know that the corrections made has been considered sufficient
My best regards
Reviewer 3 Report
Re: Correlation between caries, body mass index and occlusion in an Italian pediatric patients sample: a transverse observational study
General comments: The authors have made a reasonable effort to improve on this manuscript, however, it has some shortcomings that need to be addressed. While the scientific rigor is reasonable, the format its written, reduces its strength and robustness. In particular the way the sentences are structured across the manuscript.
Abstract
Line 26, it reads “Methods: For the study has been enrolled 127 patients (72 Female, 55 Male), aged between 6 and 16 years (mean age 10.2) related to the Department of Pediatric Dentistry, University of Messina, between January and June 2019”
Comments: Sentence need to be restructured
Line 28, it reads “Teeth examined has been coded with the DMFT index, quantitative BMI values has been grouped into 4 categories (underweight, normal weight, risk of overweight, overweight), on the basis of a table adjusted for age and gender. “
Comments: Sentence need to be restructured
Introduction:
Line 42, it reads “aged between 5 and 19 years; whit the aim to observe how the levels of body mass index (BMI) and”
Comments: Typo underlined
Line 45, it reads “Oral health and obesity share common risk factors: such as genetic, socioeconomic and food problems”
Comment: vague statement underlined. What food problems?
Comment: Consider writing the hypothesis of your study, before the aims.
Materials and Methods:
Line 75, it reads “Patients’ weight (kg) and height (cm) were registered two times for each one, in order to prevent any measurement error”
Comment: vague statement underlined. Each one? Re-write for clarity.
Line 93, it reads “During patients examination has been followed the guide lines to prevent infections using 93masks, gloves and a first visit kit composed by a mirror, a probe and a tweezers.”
Comments: Sentence need to be re-structured
Line 101, it reads “The evaluation has been repeated in 20 randomly selected patients, to assess the intra-operator reliability.”
Comments: What was the result or level of agreement?
Results:
Comments: There is total reliance on tables. There are not clear written details of the data. Some of the data are described in the discussion. They should be in the results, and the discussion left for interpretation of the data and comparison with the literature.
Discussion:
Comments: The discussion is generally difficult to read, due to many sentences that need to be structured
There are also data details that should not be in the discussion. For example, in line 240, it reads “Normal weight patients showed a significant inverse correlation of the two variables, as BMI increases, the presence of caries decreases (Table.8).”
Comments: information that relates to the results of the tables should be in the results section and not in the discussion.
There is generally correlation between BMI and caries, however, the third important component to this study which is occlusion/malocclusion has not been discussed and I believe the correlation between the three variables should feature prominently in the discussion, since this the core of this manuscript and its novelty.
There is no study without limitation/s. It is important for the author to present what they believe is the limitation/s to their study, and possibly how this can be addressed in future studies.
Author Response
Dear Reviewer
Thank you so much for your valuable time and usefull contribution. We appreciate the imputs received from you, and thank to your suggestions we have improved the manuscript entitled “Correlation between caries, body mass index and occlusion in an Italian pediatric patients sample: a transverse observational study”. I illustrate, point by point, all the corrections performed following your suggestions.
Abstract
Line 26, it reads “Methods: For the study has been enrolled 127 patients (72 Female, 55 Male), aged between 6 and 16 years (mean age 10.2) related to the Department of Pediatric Dentistry, University of Messina, between January and June 2019”
Comments: Sentence need to be restructured
Sentence has been restructured as requested.
Line 28, it reads “Teeth examined has been coded with the DMFT index, quantitative BMI values has been grouped into 4 categories (underweight, normal weight, risk of overweight, overweight), on the basis of a table adjusted for age and gender. “
Comments: Sentence need to be restructured
Sentence has been restructured as requested.
Introduction:
Line 42, it reads “aged between 5 and 19 years; whit the aim to observe how the levels of body mass index (BMI) and”
Comments: Typo underlined
Sentence has been restructured as requested.
Line 45, it reads “Oral health and obesity share common risk factors: such as genetic, socioeconomic and food problems”
Comment: vague statement underlined. What food problems?
We clarify that the risk factor is related to eating disorders
Comment: Consider writing the hypothesis of your study, before the aims.
Authors’ hypotesis has been inserted before the aim of the study
Materials and Methods:
Line 75, it reads “Patients’ weight (kg) and height (cm) were registered two times for each one, in order to prevent any measurement error”
Comment: vague statement underlined. Each one? Re-write for clarity.
We clarified the statement
Line 93, it reads “During patients examination has been followed the guide lines to prevent infections using 93masks, gloves and a first visit kit composed by a mirror, a probe and a tweezers.”
Comments: Sentence need to be re-structured
Sentence has been restructured as requested.
Line 101, it reads “The evaluation has been repeated in 20 randomly selected patients, to assess the intra-operator reliability.”
Comments: What was the result or level of agreement?
Systematic and random errors were calculated comparing the first and second measurements with dependent t-tests and Dahlberg’s formula, at a significance level of P<0.05. All measurement error coefficients were found to be adequate for appropriate reproducibility of the study
Results:
Comments: There is total reliance on tables. There are not clear written details of the data. Some of the data are described in the discussion. They should be in the results, and the discussion left for interpretation of the data and comparison with the literature.
The data reported in the discussion section have bee eliminated, leaving only an interpretation of the data itself, in comparison with litterature findings.
Discussion:
Comments: The discussion is generally difficult to read, due to many sentences that need to be structured
There are also data details that should not be in the discussion. For example, in line 240, it reads “Normal weight patients showed a significant inverse correlation of the two variables, as BMI increases, the presence of caries decreases (Table.8).”
Comments: information that relates to the results of the tables should be in the results section and not in the discussion.
There is generally correlation between BMI and caries, however, the third important component to this study which is occlusion/malocclusion has not been discussed and I believe the correlation between the three variables should feature prominently in the discussion, since this the core of this manuscript and its novelty.
There is no study without limitation/s. It is important for the author to present what they believe is the limitation/s to their study, and possibly how this can be addressed in future studies.
Discussion section has been revised, and sentences restructured, in order to be more understandable.
I hope that the corrections made to the manuscript are good for you.
My best regards